# Sustaining a Transformative Disaster Risk Reduction Strategy: Grandmothers’ Telling and Singing Tsunami Stories for over 100 Years Saving Lives on Simeulue Island

**DOI:** 10.3390/ijerph17217764

**Published:** 2020-10-23

**Authors:** Stephen A. Sutton, Douglas Paton, Petra Buergelt, Saut Sagala, Ella Meilianda

**Affiliations:** 1College of Health and Human Sciences, Charles Darwin University, Darwin 0815, Australia; douglas.paton@cdu.edu.au; 2Faculty of Health, School of Health Sciences, University of Canberra, Canberra 2617, Australia; petra.buergelt@canberra.edu.au; 3School of Architecture, Planning and Policy Development, Bandung Institute of Technology (ITB), Bandung 40132, Indonesia; saut.sagala@sappk.itb.ac.id; 4Tsunami and Disaster Mitigation Research Centre, Syiah Kuala University, Banda Aceh 23232, Indonesia; ella.meilianda@tdmrc.org

**Keywords:** natural hazards, transformation, tsunami, grandmothers, disaster risk reduction

## Abstract

As projections about the number and scale of natural hazard events and their impact on human populations grow, increasing attention is being paid to developing effective means for preparing for and mitigating those impacts. At the same time there is an emerging understanding that gradual and incremental changes in disaster risk reduction (DRR) will not adequately meet the future needs of vulnerable populations. Transformational changes have been identified as a necessary requirement to avoid ongoing large-scale losses of life and property and models have been proposed to recalibrate DRR strategies to achieve transformative changes. One cited example of a transformative change in DRR is that of Simeulue Island. Simeulue Island suffered two tsunamis approximately 100 years apart (1907, 2004) with markedly different impacts. This paper looks in detail at the cognitive and developmental mechanisms Simeulue co-opted to sustain the transformational change throughout the 20th century. Information from interviews and observation identified the role of grandmothers have in the effective communication of risk as well as motivating appropriate action to save lives. The possibility of similarly overlooked, local, and pre-existing community capacities for transformative change in DRR is then discussed.

## 1. Introduction

There is increasing concern about the extent to which human populations around the world will be vulnerable to natural hazards in the foreseeable future [1]. In addition to the predicted increases in meteorological hazards such as hurricanes and typhoons, climate change is contributing to larger and more hazardous wildfires and flooding as well as related secondary hazards such as landslides [2].

One consequence of changes in natural hazard risk is that baseline assumptions that underpin individual and organizational policy decisions are being brought into question. For example, urban and regional planning decisions made in the 20th century are already contributing to negative outcomes for communities in a 21st century characterized by higher levels of risk realization [3]; houses and residential areas constructed in zones previously considered to be at low to moderate risk are now high risk zones threatened by natural hazards. Coastal properties that were once highly sought after are washing into the sea as coastlines erode with sea-level rise [4]. The consequent DRR (Disaster Risk Reduction) implications for families, neighborhoods, and jurisdictional authorities and related impacts on mental health and wellbeing, insurance, livelihoods and economic vitality are rising in unprecedented ways.

In a recent paper, Paton and Buergelt [5] present a model that calls for the development of research designed to create transformative pre-event DRR strategies to meet these unprecedented challenges. They emphasize that a rethink is required if there is to be more than incremental improvement in DRR. They suggest that transformation needs to include shifts in thinking about “social-environmental relationships, increasing risk acceptance in the context of evolving landscapes and countering beliefs regarding not responding” [5] (p.1). They show that successful examples of long-term sustained programs of preparation for natural hazards have some form of social transformation as a foundational element. Paton and Buergelt argue that the embedding of a transformative element in DRR programs is critical to sustained resilience development.

This paper expands on Paton and Buergelt’s [5] model in two ways. First, it expands on one of the case studies they refer to, namely that of Simeulue Island in Indonesia. In 2005 the United Nations presented the people of Simeulue Island the Sasakawa Award for DRR [6] in recognition of the success of the island’s transformative DRR strategy. The successful delivery of the entire population from the *smong* (the local language term for near-field tsunami) of 2004 is made more remarkable by the facts of Simeulue’s proximity to the epicenter of the tsunamigenic earthquake and the island’s political and cultural connections with mainland Aceh, which suffered devastating losses. The island’s survival was a surprise.

In their paper, Paton and Buergelt [5] identified some potential elements that may underpin a transformative component of a DRR program including the way community leaders facilitate participation, collective efficacy, empowerment and trust, and the use of arts as transformative pathway. This paper will examine the constituents of the DRR transformation implemented on Simeulue island and demonstrate that grandmothers are community leaders who are use arts to sustainably shift the perspectives of residents in ways that has them being able to effectively recognize the signs of *smong* and respond correctly to *smong*.

Second, the paper develops the observation that for a transformation to be “contagious” (collective or social transformation) it needs to create or tap into and facilitate a schema or narrative that is widely held within a groups sense of coherence (“SOC”) [7,8]. SOC is a “global orientation that expresses the extent to which one has a pervasive, enduring, though dynamic feeling of confidence that (a) the stimuli deriving from one’s internal and external environments are structured, predictable, and explicable; (b) the resources are available to one to meet the demands posed by these stimuli; and (c) these demands are challenges, worthy of investment and engagement" [8] (p. 19). A person with a strong SOC may be more likely to successfully appraise the risk potential of an event, take on challenging situations and integrate setbacks better than someone with weakly developed SOC [9]. These characteristics are all related to an individual’s resilience [10] (p. 385) [11]. SOC incorporates an inner trust element, which “leads people to identify, benefit, use, and reuse the resources at their disposal” [12] (p. 138). The resources include social capital which is a “community level or ecological factor” [13] (p. 160) and in which trust plays a central role in any action [13] (p. 166).

In appropriate circumstances, “it may be possible for different cultures to have their own, culturally relevant translations for SOC so that it becomes a meaningful protective factor when confronting stressful situations” [14]. Thus, creating or tapping-into and facilitating “culturally relevant” components of SOC might offer the potential to enable a DRR program to become self-sustaining.

It will be argued that, consistent with Paton and Buergelt’s model [5], the successful transformation of Simeulue’s approach to DRR during the 20th century pivoted on a deep trust in the community’s SOC. This observation may offer important implications for developing DRR strategies with a transformative component capable of leading to long-term sustainability in other contexts. We set out the example of Simeulue in some detail and look at the results of an ethnographic research study of narrative interviews with survivors of the 2004 Indian Ocean tsunami. Key factors identified in the research are then examined to understand the socio-cognitive reasons those factors were effective. The primary role of grandmothers in transmitting DRR information is considered in the context of an emerging understanding of their role in human development and cognition. Six principles which appear to underpin the transformative DRR program in Simeulue are then described and it is proposed that these features are embedded within society, such that the sort of transformative strategies required for sustained DRR may already exist in most societies.

### Case Study Area: Simeulue Island and “Smong”

Simeulue is an island about 100km long and the most northerly in the chain of islands along the west coast of Sumatra (see Figure 1). Originally a part of the Dutch East Indies, it was incorporated into the new nation of Indonesia in 1945 and is part of Aceh Province. During the Dutch colonial period, on the afternoon of Friday, 4 January 1907 a M7.6 earthquake struck off the coast of Simeulue with a subsequent tsunami that “devastated Simeulue and extended over 950km along the coast…” of mainland Sumatra [15] (p. 427). (Afternoon is local time). There is some question about the magnitude of the earthquake, given the scale of associated disruption and tsunami (Newcomb et. al 1987). The Orang Simeulue (the people of Simeulue) called it a “*smong*”–in fact, for most of the 20th century it was “*The smong*”. (“Orang Simeulue” express a preference for the use of “smong” rather than tsunami and the term will be used throughout the rest of the paper). The word *smong* means “tsunami” and existed in the local languages of Simeulue before 1907 but does not appear to have been connected to any DRR wisdom at that time [16]. 

Nearly 100 years later, this time on a Sunday morning, on 26 December 2004, another earthquake struck Simeulue [17]. The 9.2 Mw earthquake epicenter was less than 50km off the coast and triggered a tsunami that struck Simeulue shortly after. But this time, and in stark contrast to the previous *smong*’s impact and despite population growth to nearly 80,000, only seven deaths were reported for the island [18].

Early research into the low death toll, which is also striking given the estimated 230,000 deaths along the coast of mainland Aceh [19], hinted that between 1907 and 2004 Orang Simeulue had transformed what they understood about *smong*, specifically, what they needed to do to survive it. The rapid and successful evacuation of the entire coast of the island was attributed to a *smong* narrative initiated by the survivors in 1907 which had remained in circulation among Orang Simeulue ever since [16,20,21]. The story sets out the signs of an impending *smong*, describing how a huge earthquake results in the recession of the sea and exhorting the listener to run, without delay, to the mountains [22]. The story has been identified as an example of traditional knowledge that saved lives [23,24] and it is assumed that it is the story *per se* that was the critical component of Simeulue’s DRR success [17,20]. However, research shows that knowledge of hazard risk does not usually lead to transformative change in DRR preparation [25,26].

For example, like Simeulue, Banda Aceh had a unique linguistic term for tsunami, but there was no widespread sharing of the wisdom of “*Ie beuna”* in a DRR context [21]. Ayi, a local from Banda Aceh stated that he had “never heard stories from my parents or grandparents related to tsunami before 2004” [27] (p. 111). The absence of the *smong* story was also the case in Simeulue prior to 1907 [16] The word “*smong*” predates the 1907 event and was only subsequently linked in a transformative way through a narrative communicated to children, initially by mothers and then by grandmothers throughout the island.

In another instance, the traditional knowledge and practices relating to the Merapi volcano may have contributed to unwillingness to evacuate [28,29] while in some tsunami-affected areas of Japan historic warning markers were ignored [30]. 

## 2. Materials and Methods 

The current research sought to investigate the socio-cultural mechanisms that were implemented by Orang Simeulue that facilitated the *smong* narrative supporting every individual on the island people identifying that a tsunami was occurring and empowering everybody to take immediate, appropriate action. This inquiry was informed by two observations about DRR narratives. The first is that narratives, traditional or otherwise, are not always effective in enabling people to recognize when extreme natural processes occur and how to respond to them appropriately (above), and therefore close attention should be paid to those that do work, like Simeulue. Identification of the factors including content, context and mode of transmission that amplified the effectiveness of the Simeulue narrative is important as this knowledge has the potential to inform the development of transformative and sustainable DRR strategies in other situations. 

The second observation is that in educated, industrialized, rich and democratic (“WEIRD”) Western societies [31] have not been able to achieve sustained, long-term and widespread transmission of hard-won disaster knowledge [25,32,33]. Paton and Buergelt [5] state that people in Western societies commonly hold a philosophical worldview that includes people believing that they are separate from and superior to the environment, which leads to the anthropocentric belief that the environment is a resource to be controlled and exploited. This worldview also leads to citizens being “educated” in ways that perpetuate Western cultures and weakens and oppresses citizens [34], and citizens transferring responsibility for the control of natural hazards to government agencies and authorities. Importantly, western education pedagogies and transfer of risk responsibility result in narratives of DRR practice and experience, apart from entertainment value, failing to trigger sustained interest and being dismissed as irrelevant to the daily concerns of the individual. There is a need to identify effective transformative programs that incorporate new narratives as core or part of their practice [35,36].

WEIRD cultures are, however, more unusual than they think. As Geertz [37] (p. 48) puts it: “The Western conception of the person as a bounded, unique, more or less integrated motivational and cognitive universe, a dynamic center of awareness, emotion, judgment, and action organized into a distinctive whole and set contrastively both against other such wholes and against a social and natural background is, however incorrigible it may seem to us, a rather peculiar idea within the context of the world’s cultures.” These views are absorbed over time and are “predominately learned through informal interactions with other people (e.g., education, television, magazines, music, internet)” [34] (p. 3). In other, less peculiar world views, the informal interactions identify that responsibility for risk is not subject to anthropocentric partitioning because there is “ontological understanding that the land and people are one symbiotic unit” [38] (p. 2). In the context of traditional societies, both the reality of daily life and ontological perspectives cause individuals to be closer to natural processes than WEIRD societies, and this proximity may inspire ongoing attention to narratives about natural hazard risks. While Indigenous ecological knowledge has, until recently, only been transmitted as oral narrative [39] it has been shown that it can sustain risk communication over long periods of time [40].

In order to investigate the DRR narrative tradition of Simeulue the research used a synergy of structuralist perspective [41], narrative theory [42,43,44], and the ecological all-hazard inter-disciplinary risk management and adaptation model [45] as a philosophical and topic-specific lens for focusing cultural elements of Simeulue’s DRR. To obtain a “crystallized view” [46] (p. 963) a series of narrative interviews were conducted with Orang Simeulue, who had survived the *smong* in 2004. The ethnography was conducted over 3.5 months between 2016 and 2017 at various locations around the island (see Figure 2). Interviews were conducted in coffee shops, family homes and yards, sporting fixtures and, in one case, a gaol. Efforts were made to obtain a diverse sample of the population in terms of occupations, gender and geographic context. Virtually all Orang Simeulue live on the narrow strip of relatively flat land around the coastal margins of the island. 

All adults encountered during the fieldwork who were there in 2004, regardless of whether formally interviewed or not, had a lived experience of the *smong*, total of 37 narrative interviews with 58 people were conducted. The research participants include a mix of urban and rural men and women over a wide age range (see Table 1). Most of those interviewed were quite young at the time of the *smong*, the youngest being six the oldest was 80+. To enable participants to fully express themselves, interviews were conducted in a mix of Indonesian and Acehnese with the assistance of an interpreter from the region with qualifications and experience in research and natural hazards. With local cultural knowledge, the interpreter was able to conduct both “on-the-fly” interpretations during the interview and re-interpretation during digital transcription, which was conducted in the field, usually within 24 hours of interview.

## 3. Results

A key feature of the testimony of those interviewed across all age ranges was a story about “*smong*”. Early in the project it became clear to research team that grandmothers played an important role in the DRR story. So much so that during the last interview of the first fieldwork season both interviewer and interpreter were surprised when, more than 20 minutes into the story, the participant had not made mention of grandmothers at all. When he subsequently mentioned that he was an orphan the situation became clear and his exceptional reticence seemed to “prove the rule” of grandmothers in Simeulue’s DRR transformation. (Participant D was interviewed in Matanurung village on 15 December 2016).

Throughout the interviews, participants attributed their survival to the DRR wisdom of their grandmothers (see below). Orang Simeulue related how they had received the story when they were children. The story, which had come from their ancestors, was passed on to them most frequently by their grandmother. Sixty percent of interviewees referred specifically to grandmothers while other elders such as grandfathers (31%) and “grandparents” in general (31%) were less frequently cited as the source of the DRR information. A word frequency count of elder terms from the entire corpus of transcribed interviews identified similar results with “grandmother” comprising 56% of while “grandfather” made up only 13% with “grandparents” 11% and “ancestors” 19%. These figures are not mutually exclusive, many of those interviewed referred to grandparents and ancestors as well as grandmothers. However, these numbers reflect the observations of the research team in the field that grandmothers are the most important category of DRR informant in Simeulue.

The raw numbers derived from counts of references in interviews indicate that more than half of those who participated in the research attribute their knowledge (and consequent survival) to their grandmother. However, the numbers alone don’t convey the emotional connection many of those interviewed expressed when talking about their grandmother.

Participant B for example stated:


*My grandmother told me about smong. The role of Grandma in my period is like, she always spoils her grandchildren, so we are closer to our Grandma than our parents. They put us to bed, they give us money...so we are closer to them than to our parents. So, before they put us to bed, or while sitting and relaxing they tell us that if someday the big earthquake happens usually a big wave will come...*
*(*
*Participant B was interviewed at Suka Jaya village on 9 December 2016).*


Participant C also said that he had a special relationship with his grandmother:


*I knew the story because I am the youngest and I was very close to my grandmother. She was a hero for me. Because, when I was close to her, my parents, my siblings didn’t dare to disturb me [laugh]. Because I was close to her, I often gave her messages, so she liked to share her experience.(*
*Participant C was interviewed at Matanurung village on 14 March 2017).*


Participant A was quite animated as he related stories about his grandmother:


*The earthquake happened, my grandmother said. Her skin was wrinkled but her hair was still black, and her teeth were good. She said "If earthquake happens run, don’t bring anything. After the earthquake take the rice, the water, clothes, trousers and check the sea, if the water recedes “RUN FAR” she said – shaking her finger. (*
*Participant A was interviewed near Abail village on 12 December 2016).*


Participant A went on to emphasize his grandmother’s passion and consistency in reprising the *smong* the story every Thursday evening as the family gathered for the semi-formal reading of the Koran in preparation for Friday prayers at the mosque. As he grew older, he became bored with the story:


*I was small I thought “it’s a story which people take for granted.” So, on Thursday night I just think “oh there’s the story of earthquake again, it’s always the same story!”*


Participant A recalled that after being thrown to the ground by the massive earthquake in 2004 he prayed for forgiveness from his grandmother who had died in 2002. 

The story Participant A’s grandmother told him was consistent across those interviewed: 


*In 1907 there was a large earthquake followed by a smong. Nearly everyone was killed. So, if there is a big earthquake and the sea recedes – run to the mountain. Don’t wait just run.*


Participant L stated that he got information:


*...from our ancestors. They teach us, give information to us that if an earthquake happens, we have to go to the mountains. By preparing kids by singing songs at bedtime with lyrics about running to the mountain after a big earthquake. The lyrics remind us not to go down to pick up fish when the sea recedes, just save yourself. Run to a higher place when the earthquake comes. (*
*Participant L was interviewed at Suak Buluk on 7 December 2016).*


For Orang Simeulue, ancestors are conflated with grandparents, and all are revered [47]. Within families, details were included in the story about relatives who had died at the time. This made the story specific and relevant, adding personal connection to the key messages grandmothers were giving to their families across the island.

The story was told first to infants and sung in lullabies at bed-ime. For example, Sarman shared:


*…got the story of smong from my parents and grandparents when I was little, before I went to sleep. (*
*Participant E, interviewed at Suka Jaya 4 December 2016).*


Yogaswara & Yulianto [48] have noted that the *smong* lullaby does not have a set lyrical format, but is a loose and extemporized musical telling of the story. A third component of the DRR communication was musical; the traditional lament of Simeulue known as “*nandong”*. Again, this medium has strong emotional overtones with the music characteristically mournfully sad. Often sung by a grandmother with only a “*gedung*” drum as an accompaniment, the *nandong* is a regular feature at formal occasions such as weddings but is also performed in more impromptu settings in the round of village life. This musical reinforcement of the risk communication has important mnemonic and motivational implications for the individual [49,50].

In the same way that nandong is an adult musical construction of the *smong* lullaby, an “adult” version of the *smong* story (“*inafi-nafi*”) was told in a range of family and village settings. These stories incorporated greater detail (see below) but were always consistent with the short version of the story above. When telling the story, grandmothers or other older community members would often become quite animated, yelling and exhorting the listeners to run should a *smong* occur.

## 4. Discussion

### 4.1. The 1907 Tsunami Catalyst for Grandmothers Leading Transformative DRR 

It is expected that people learn from disasters. Sattler et al. [51] (p. 1397) state, for instance, that: “someone who survives a hurricane may gain knowledge about the potential for destruction; the value and benefits of preparation and evacuation; how to recover in its aftermath; or develop an enhanced sense of self-efficacy, new skills, and new ways of coping with subsequent disaster threats. Furthermore, someone who has taken actions to minimize damage, secure his or her family’s safety, or replace lost or damaged possessions may develop an enhanced sense of self-esteem, self-efficacy (personal characteristic resources), and stronger bonds among family and community members.” Efforts to institutionalize learning from disasters are common, for example, the Victorian Bushfires Royal Commission sought “modest and targeted organizational reform as a catalyst for change” [52] (p. 18). The Royal Commission’s focus here was on incremental “top-down” changes within government agencies to “improve operational performance”. These lessons may lead to institutional change however the target has not been transformative change.

Research indicates that efforts to impose control on hazards through the increasingly complex and top-down creation and documentation of disaster management plans [53] may have “perverse” consequences potential consequences including low levels of disaster preparation [54]. The formal planning processes may, in fact, disempower individuals and communities who then place greater reliance on government or “someone else” to warn them of, and rescue them from, natural hazard events [55,56,57]. In contrast research indicates that, to be effective, rather than “top down” DRR behaviors need to be embedded into the normal everyday practices of a community [25,45,55]. There is evidence that embedded cultural understandings of the environment and hazards together with life-long learning are effective contributors to “resilience” [34,58,59,60] and where there is a diminished sense of connection with “community”, individuals and families become relatively more vulnerable [61,62,63,64]. Instead, a focus on community leadership and support systems is essential for long-term sustainability in DRR action [25,57].

Paton and Buergelt [5] present some examples where a disaster was a catalyst for a transformative change in DRR behaviors. Their point is that a disaster can catalyze the *transformation* in how people interacts with a natural hazard. They note that “the relationship between a disaster (the catalyst) and new, transformative ways of thinking and acting is mediated by social factors such as community leadership, active community participation, collective efficacy, sense of community, social identity, and trust” [5] (p. 11).

Simeulue Island presents one example; in this case, the 1907 *smong* event was the catalyst for the sustained transformative DRR action. The leadership of grandmothers telling and singing the *smong* story in Simeulue for the rest of the 20th century was central to that transformation. Interview informants consistently refer to the date of the event (57% of locals interviewed specifically mention the year 1907) and describe how their surviving ancestors lived in the hills for a long time, many months, fearing aftershocks which might trigger further *smong*. When they returned to the places their villages had been, they believed they were the only survivors and were pleased to find some others had also been lucky. These were the people who initiated the transformation in risk communication on Simeulue.

Of course, the individuals we refer to as grandmothers were at one time daughters and mothers. At some time after 1907, they were the survivors or the daughters of survivors of the *smong*. For them the trauma of the event was real, including the personal narratives of parents and their own experience. Even for those who did not see, feel, hear, smell or taste the *smong*, its impacts would have been everywhere to see for many years after the event. Denuded villages, dead fields and debris piled up against the hills would have remained visible for as much as a generation after the *smong* event, bringing an immediacy to transmissions of the saga through story and song.

While we can only speculate on the trauma those survivors suffered and how this trauma transformed into the *smong* awareness practices they initiated, one hundred years later the practices were quite standardized across the island’s 80,000 people. Prior to 2004, most of the inhabitants of Simeulue had received information about *smong* in three ways; stories, lullabies and traditional songs called “*nandong*”. In each case the information was transmitted by family and village elders, most notably grandmothers.

### 4.2. Consistent Simeulue DRR Stories

While the core elements of the stories are consistent, specific details about family members killed or saved may be included from time to time. Shorter versions were more perfunctory, but all contained the same message: “If there is a very big earthquake, and the sea recedes, run to the mountain; don’t wait – just run!”. The nandong contains the same message, albeit with some poetic accompaniment [16]. Longer forms of the story detailed the deaths and lucky escapes of close relatives as well as natural phenomena like “sharks caught in trees”. “The result of…” was a universally trusted risk communication. Within minutes of the violence of the earthquake ending, the entire population of Simeulue was running for the nearby hills. Some, usually young men, were allocated to check the nearby ocean to see if it had receded. It had. One informant described his shock that the sea had not merely receded as it does on the low tide but was “gone”! “*There was no water all the way out to the (nearby) island. It was just gone!*” (Participant F, interviewed at Salang, 8 December 2016). Within a minute or two those checking the sea levels were back exhorting everyone “*Smong! Smong! Lari! Lari*! (Run! Run!)”.

### 4.3. Singing for Emphasis of the Simeulue DRR Message

Tehub et al [65] describe how lullabies, as well as providing a crooning comfort to infants, are also a mechanism for mothers to assuage their mental anguish. Quietly singing about their grief and their fears in a universally recognized lullaby format [66,67], survivors started a cultural tradition that was inculcated to subsequent generations as the population grew from the small survivor group. Those surviving mothers reprised their duty of care when the time came by singing lullabies to their children’s children, reinforcing both the message and the practice such that it is a well-recognized trope in Simeulue today [68]. The importance of the narrative was not merely to provide information to listeners, but also to assist with the integration of the *smong* into the sense of coherence of the surviving narrator; “We organize our experience and memory of human happenings mainly in the form of narrative..." [69] (p. 4). Narratives are distinguished by their ability to compress and encode a great deal of information about the world, including the causal relations between events over time [70]. “We each seek to provide our scattered and often confusing experiences with a sense of coherence by arranging the episodes of our lives into stories” [71] (p. 11). Indeed, the maintenance of ongoing mental health is related to our capacity to sustain a narrative which incorporates these events and the change the cause [72,73,74]. In time, the message of the lullabies was complemented with the emergence of the broader *smong* narrative. It does not require any great speculation to perceive survivors sharing their experiences with each other and with their families as time went by. Despite the tumult of the 20th century, the 1907 *smong* remained the biggest event in their lives; overshadowing all else in its scale but especially in terms of the grief and suffering it created.

An additional strength of the Simeulue *smong* strategy may lie in the transmission of the stories and songs by grandmothers to children at bedtime. There is compelling evidence that learning prior to sleep is very effective in establishing firm memories [75,76,77]. Not only were infants and children taught the story by a trusted and admired individual, they were presented the details at their most receptive, almost guaranteeing a life-long remembrance of the essential components of DRR. At first, the singing of lullabies about the 1907 disaster may have helped calm surviving mothers and babies, but the tradition that was created aided the meaningful integration of the *smong* event into the sense of coherence of both mothers and their babies and thus subsequent generations.

The *nandong* can only be considered in a similar light, albeit with the added depth of transmission of emotional cues [78]. While the story is inherently emotional, combining lyrics of hazard risk and death with the sad musical tones of the *nandong* serves to add an additional layer to the several already affected by an individual’s knowledge of the story from their infancy [16]. In each “phase” of communication an individual’s sense of coherence is being tuned and enhanced such that the knowledge of *smong* becomes ontological and avoidance behavior becomes habituated. In Simeulue, the starting point for this habituation is grandmothers.

### 4.4. Grandmothers as a (Re)source for Transformation to Sustainable DRR

Grandmothers “operate” at a different level to agencies, advocates and even parents. In Simeulue, as elsewhere, grandmothers are an intimately trusted part of the family and *kampung* and, for grandchildren, they are the ultimate insiders. (Kampung translates as “village” but more generally refers to the local interacting, and usually inter-related, resident group). Typically, grandchildren accept their presence and input as wholly positive and solely focused on their wellbeing. As individuals emerge into adulthood the emotional bonds linked to the reality and memories associated with grandmothers remain. These memories are often foundational anchors in an individual’s development will be tethered to the individual’s Sense of Coherence. This profound developmental role for grandmothers in the life of the individual may reflect their role in the development of the species. Observations of the health benefits that accrue to grandchildren in families where the grandmother is an active contributor suggest an adaptive advantage that may have played a significant role in the survival of humanity [79,80]. The Grandmother Hypothesis [81,82] draws together these and other observations about human biology and society that point to a pivotal role for grandmothers. The chief component of the Grandmother Hypothesis is longevity. Among the primates, humans are the only species that include an extensive post-fertile life stage. Remaining alive and active after menopause enabled human grandmothers to cook for and aid their daughters in ways that contributed to the health status of both daughters and grandchildren. According to Hawkes, grandmother longevity and their contributions to their descendants’ diet and health has provided more than a mere spandrel life extension for males, “… grandmothering drove the evolution of genus *Homo*” [83].

Within this framework, the evolution of a distinctly human infant prosocial capacity to emotionally engage with multiple caregivers (e.g., including grandmothers as well as mothers) is argued by Hawkes to be a key contributor to the development of our unique psychology [83] (p. 299). These prosocial capacities become particularly important for the survival of a child after weaning, especially after the birth of a sibling, when a mother’s energy and priorities shift to the new infant and the support of a grandmother makes a critical difference. Hawkes [84] argues that the relationships engendered between infants and grandmothers is the evolutionary foundation of human social cognition. Unlike other primates, human infants develop one-to-many caring relationships with grandmothers in a primary place as provider and supporter. This relationship capacity forms the basis of human preferences for participation, engagement, and sharing of attention and intentions [84] (p. 5). The attachments formed in infancy are internalized as “working models of attachment” [85] (p. 58); stable patterns of cognitions and behaviours supportive of relationships founded in security and trust. These relationship capacities are proclivities in all humans and in some cases the pro-social traits are entrained or codified into formal and semi-formal social structures which foster involvement by all age groups. Examples of healthy formalized intergenerational interactions include “*gotong royong*” in Indonesia [86] and “*chonaikai”* in Japan [87,88] (*Gotong royong* translates as “mutual help” or “mutual assistance”, *Chonaikai* translates as “community council” and refers to local neighbourhood governance arrangements). These organizations conduct DRR programs (among other things) while providing ongoing opportunities for intergenerational interactions and thus transfer of skills and knowledge which recapitulate traditional ways of learning [89,90,91].

In Simeulue, the developmental features of the relationships and behaviours relating to *smong* reflect the contribution of grandmothers over a lifetime to coalesce an understanding of tsunami risk and an empowering motivation as part of Orang Simeulue’s SOC. “Grandmothering” is the critical capacity for initiating and sustaining the transformation which saw inculcation of risk knowledge and actions, and their habituation, of the entire population. This grandmothering “capacity” existed in Simuelue before it was co-opted as a DRR strategy, and indeed probably exists in all populations.

Szinovacz [92] states that the lived experience of grandmothers, either having one or being one, is near-universal. So much so that there is a potential that it is taken for granted. In WEIRD societies, demographic and economic changes since the industrial revolution have significantly altered the customary roles of grandparents [93,94]. While the majority of humans, through time and space, include grandmothers as an integral quotidian component of lifeways, WEIRD societies developed a tendency to separate the nuclear family from grandparents and ultimately to separate off grandparents into geriatric enclaves. However, research indicates that even in WEIRD societies, such as the United States, grandparents remain a critical resource for raising grandchildren and for maintaining family social and economic responsibilities, particularly in times of resource stress (Livingston et. al 2010). Cultural differences do exist in the ways grandparents engage with families and grandchildren, but the reliability of the interaction appears to be cross-cultural [95]. That is, even in societies where “taking grandmother for granted” may lead to placing her in aged care, it also means relying on her help when times are tough.

In sum, while the nature of change in the role of grandmothers is temporally and geographically variable, everywhere they remain an important resource and most trusted part of the social fabric. It appears that the contribution of grandmothers over the course of human development has had a transformative effect on both the biological and cultural features of all societies. In Simeulue, we have a contemporary example of how that transformative power has been applied to DRR. 

### 4.5. Simeulue’s Transformative DRR Strategy

While the importance of the DRR narrative has been noted by others [20], the transformative role performed by establishing grandmothers as custodians and narrators of the DRR wisdom has been overlooked, perhaps due to WEIRD subjectivity (the authors may also have missed this point due to their subjective WEIRD positionality, had it not been for the repeated passionate testimony of the people of Simeulue).

In order to become clear about the significance of the success of Simeulue’s DRR, some discussion is warranted of the physiognomic, physical, and cognitive hurdles that were overcome to achieve the evacuation.

To appreciate the effectiveness of the Simeulue DRR transformation, it is important to take into account what people were experiencing at the time. On an island that experiences a high number of earthquakes, the Great Earthquake of 2004 was unprecedented; it being 1000 times stronger than the relatively common 6.0 Mw quakes and even 100 times stronger than a 7.2 Mw quake which occurred in 2002. Birds and animals were behaving strangely just prior to the quake. The sound and violence threw people to the ground, leaving them bruised, lacerated and nauseous. Many thought it was “doomsday”; the day of judgement and the end of the world. They felt “panicked” and thought they were about to die. The 9.2 Mw caused many survivors to report a strong physiognomic reaction in the form of dizziness and nausea. When the shaking stopped, people were lying on the ground (it had proved impossible to stand) and felt sick and disoriented. This phenomenon has been observed elsewhere and is generally considered to be a by-product of infrasound waves released by the earthquake [96]. They were experiencing an event that outstripped the imagination and seemed to pose the real threat of imminent death, creating the ultimate scenario for making “judgements under uncertainty” [97]. Despite this, they all “pulled themselves together” and took evasive action.

The physical barriers included damaged and broken infrastructure such as houses and bridges as well as the fact that the hills (“mountains”) while close were steep and paths were not necessarily well maintained.

In most natural hazard contexts, cognitive hurdles form the greatest impediment to appropriate preparation and response [25]. Efforts by advocates and governments to stimulate preparation, including information about the likelihood, timing, and impact of risk as well as practical and effective ways individuals and communities may prepare have done little to improve a generally low level of DRR action [5]. Research into human decision-making, choice and motivation around DRR and preparedness has indicated that cognitive processing is subject to a range of biases, which in normal circumstances, may be effective elisions of unnecessary and time-consuming complexity [98], but which can lead to catastrophic poor decisions in situations of uncertainty [97,99]. There are few phenomena encountered by humans that convey real situations of risk and uncertainty of the scale and gravity of natural hazards. Consequently, questions about DRR preparedness have found diverse cognitive biases interacting with aspects of personality and socio-cultural context to militate against DRR action [25].

That the evacuation following the earthquake in Simeulue in 2004 was so immediate and unanimous despite the “hurdles” indicates that the Simeulue DRR strategy had effectively overcome all of those diverse interacting cognitive barriers that contribute to fatalities in most other disasters [100]. The people on Simeulue recognizing the tsunami and responding correctly is in stark contrast to the experience of Simeulue locals who experienced the *smong* in Banda Aceh and whose yelled warnings were derided [16]. It is also in contrast to the behavior captured in videos of other recent tsunamis where people react only at the last minute (as the wave approaches) [101].

Taken together, these observations suggest that the Simeulue DRR strategy was transformative in that it empowered individuals in 2004 to either avoid or discount inertia, denial, procrastination, fear, fatalism, and negative outcome expectancy. Tasked with rapidly integrating a colossal event into their SOC and assessing what action to take, virtually every individual in Simeulue resolved to move in the same direction: up the mountain!

Using the results and the cognitive and developmental factors discussed above we can draw six narrative principles, which, taken together, formed the effective system for communicating DRR information in Simeulue. These six strategy components are presented below, together with a relevant quote from interviews illustrating each feature:An intimate, positive and supportive emotional bond between grandmothers and grandchildren (including **trust** and admiration) is the primary social context for the *smong* DRR story;○She [Grandma] was a hero for me.○*The role of Grandma in my period is like, she always spoils her grandchildren, so we are closer to our Grandma than our parents**(**Participant B interviewed at Suka Jaya on 9 December 2016).*The early transmission of DRR wisdom;○I was a kid the first time I knew about smong (Participant G, interviewed in Sinabang 3 December 2016).○*I knew about smong since I was a kid**(Participant E, interviewed at Suka Jaya 4 December 2016).*○*I was still little, so that is how I knew about that [smong] (**Participant H, interviewed in Sinabang 22 July 2017).*The persistent or sustained transmission of DRR wisdom;○*Yeah, she told the story often! (Participant I, interviewed in Langi 12 August 2017)*.○Yeah of course. They often discuss [smong]. Especially at that time we discuss if the water is coming what do we do! (Participant J, interviewed in Busung 27 July 2017).The consistency of the DRR wisdom transmitted;○*…oh there’s the story of earthquake again, it’s always the same story! (**Participant A, interviewed at Abail, 12 December 2016).*○*We know that after earthquake we have to see if the sea recedes or not. If it recedes, we have to run to the mountain as fast as possible. We know this because my grandmother said her brother died from smong. (**Participant K, interviewed in Sinabang, 31 July 2017).*○*If there is a strong earthquake, followed by the lowering of seawater, Hurry to find a place, A higher place, This is called Smong, A story of our ancestors**. (Nandong* lyrics).The use of different communication media to transmit the same DRR wisdom over a lifetime tailored to different ages;○Lullabies, stories and *nandong*The use of non-verbal communication, prosody and pitch to imbue *smong* wisdom with emotion (including fear and urgency).○“RUN FAR” she said – shaking her finger. (Participant A, interviewed at Abail, 12 December 2016).

The analysis suggests that these communication practices, which were initiated in infancy were instrumental in building the community SOC of Simeulue. Starting with a base of intimate trust and admiration, the grandmothers have used stories and song to weave DRR wisdom into the SOC of the entire community and have been doing so, now, for over 100 years. This practice started after the disaster of 1907 [16] and continued for the next 97 years.

Paton and Buergelt [5] argue that a fundamental shift or transformation in people’s thinking is required if there is to be a transformation in societal approaches to future natural hazards [102,103]. This shift must start with rethinking “how preparedness is conceptualized and facilitated through transformative learning” [5] (p. 16). They present a model based on Community Engagement Theory (“CET”) that is integrating the constructs implicated in achieving transformative outcomes in hazard preparedness [5] (p. 14). The central generative axis of the model integrates the relationship between community leadership and four factors documented as critical for DRR preparation: community participation, collective efficacy, empowerment and trust [104]. This generative axis is provoked by a catalyst such as a disaster and supported by resilient community characteristics.

The more detailed consideration of the Simeulue case study further supports the model albeit emphasizing some aspects over others. The trust and empowerment that flows from the ongoing dissemination of risk communication over a lifetime by the grandmothers of Simeulue supports observations of the importance of trust in a range of fields including DRR [13,105,106,107,108,109]. Grandmothers worked assiduously to empower their descendants to survive the next tsunami by effectively using storytelling and songs as transformative learning pathways for instilling within them the knowledge and motivation to act with “collaboration, cooperation and trust in confronting novel circumstances” [5] (p. 6).

The catalyst for the transformation that led to the directed action of the grandmothers of Simeulue was the disastrous *smong* of 1907. It can be speculated that the self-reliance and place attachment of the remote people of Simeulue contributed to the “locking in” of the elder-driven risk communication strategy that was not seen in nearby parts of the Indonesian mainland in 2004 (with devastating results).

The fact that everyone on Simeulue made the correct judgement emphasizes their trust in the information they held about what action to take. Earle [110] points out the relationship between trust and conscious consideration. If the source is highly trusted information is scrutinized less deeply while an untrusted source will have everything they say be weighed and considered. Deliberation over the veracity of data from an untrusted source can be detrimental to the effective action required in an emergency where timely responses are critical [25] (p. 148) [111].

The strength of the Simeulue strategy lies in its co-option of pre-existing cognitive and socio-cultural frameworks for the construction of SOC out of a challenging and catalytic experience. The grandmothers of Simeulue articulated risk information and self-efficacy in a climate of trust and admiration and in a way that supports a CET model of transformative and sustained DRR. Taleb [112] (p. 220) says that “to be contagious a mental category must agree with our nature.” Consistency is an important factor in human decision making [113] and a critical factor in determining the assimilation or rejection of new information [114,115,116] and an important feature of personal narratives [72,73,117]. Adger identifies the need for alignment of values attached to an individual’s world view for there to be any “practical implementation of broader adaptation activities” [118] (p. 345).

Mulyasari and Shaw [111] (p. 247) note that whether or not a community places trust in authorities hinges on whether they trust institutional actions to be based on the same values as they hold themselves [119]. While research into what these values [118] or world views [116]( may inform better construction of DRR messaging, one lesson from Simeulue is that the means of achieving narrative integration and consistency may exist “right under our noses” in cognitive and socio-cultural forms that have, to date, been largely overlooked due to being so imbedded and simple and thus “taken for granted”. This is the nurturing, consistent and continuous developmental role of trusted and admired leaders in the form of grandmothers. Grandmothers were instrumental in establishing a core DRR belief in Orang Simeulue’s SOC.

Narrative is also a fundamental human developmental construct and one that is increasingly recognized as a necessary component of effective communication. Implementing a transformative DRR strategy will not succeed without a narrative that ensures the strategy “agrees with our nature” and integrates DRR concepts and actions with our SOC. In Simeulue, the *smong* narrative was woven into the SOC of the community from an early age, bolstered by the existence of a local language term for the tsunami natural hazard and told by the most trusted of family members [16]. The Simeulue case study reinforces the idea championed by Buergelt and Paton [5] that when seeking mechanisms to create sustainable DRR transformations it is not necessary to seek new forms of human engagement. Rather, it is possible and perhaps even necessary to co-opt and revive forms of interaction that have been part of all human societies and lifeways for most of our species development.

Formalized notions of “shared responsibility” [120] echo the cooperative social contexts in human development indicated by Hawkes [84], but effective development of resilient communities based on these has proven elusive [121]. While relatively recent cultural shifts, particularly in WEIRD societies, have diminished the focus on collective shared responsibility and social identity, it is precisely these elements that have an inherent capacity to create SOC. Other forms of interaction may also be identified through consideration of other widespread social practices. For example, the commensalism common in many forms of religious practice brings people together in structured ways that suppress agonistic behavior and build trust among strangers [122,123]. To move beyond the (small) incremental shifts that have been possible within contemporary traditions of resilience development a comprehensive DRR strategy that facilitates a strengthening and expanding these pre-existing cognitive and socio-cultural frameworks is required.

Further research into the role of grandmothers play in developing Sense of Coherence around DRR or other salutogenic processes may deepen our understanding of the transformative capacity of communications based on “basic trust” [124]. One aspect of such research might consider shifts in power relationships (such as the relegation of grandmothers from leadership roles) and the ability to effectively share responsibility for DRR across different community sectors. What is the relationship between existing perceptions of power relationships and the extent to which authority is seen to be responsible for DRR and can be trusted [125,126,127,128]? Future research might also examine trust asymmetry and its links to existing perceptions, as expressed in informal narratives [129,130]. One construction that might be placed on this research is “getting to really know your local community”.

Research might also seek to identify other pre-existing socio-cultural settings that lend themselves to supporting transformative DRR practices. Examination of narratives in other cultural contexts incorporating the six narrative elements identified above may highlight forms of risk communication that are consistent with a society’s deeper “nature”. Enquiry into local narratives of ontological beliefs and shared values carries the prospect of identifying platforms for developing new and “contagious” risk communication.

## 5. Conclusions

Paton and Buergelt [5] (p. 15) call for the inclusion of community elders in DRR governance and decision making. We further extend this call to transform the context of elders in governance to a devolution of the locus of communication from merely “elders’ to grandmothers and grandfathers”. That is to say, it is not merely the “recognized” or high-status individuals easily identified as the “community elders” but also to the “everyday leaders” of extended families whose passion for their families to thrive constitutes a powerful agency for sustainable DRR.

Additionally, our findings suggest two other deeply ingrained and overlooked cognitive and socio-cultural forms that also exist as a fertile basis for the development of transformative; narrative and music. Music has been argued to have a deep-rooted base in human cognitive and linguistic development [131,132] and while today it is recognized primarily for its role in attenuating people’s mood [133,134], it is also important for its impacts on memory and motivation [135,136]. Music is also recognized as a major contributor in the use of arts in transformative learning practices [110,137].

The transformative DRR strategy in the local community of Simeulue incorporated a suite of factors that constitute effective risk communication; consistent repeated messages with accurate risk assessment and empowering calls to action. But more importantly the strategy co-opted an ancient human framework for integrating complex and difficult concepts with personal narrative and music. Establishing grandmothers as custodians and narrators of the Simeulue *smong* narrative provided the transformative step-change underpinned by feelings of personal empowerment in the face of a *smong* and strong social cohesion [138,139,140]. The success of Simeulue has its source in fundamental human activities that were already in place when they were implemented in the decades following 1907 and which remain in place in most societies around the world today.

## Figures and Tables

**Figure 1 ijerph-17-07764-f001:**
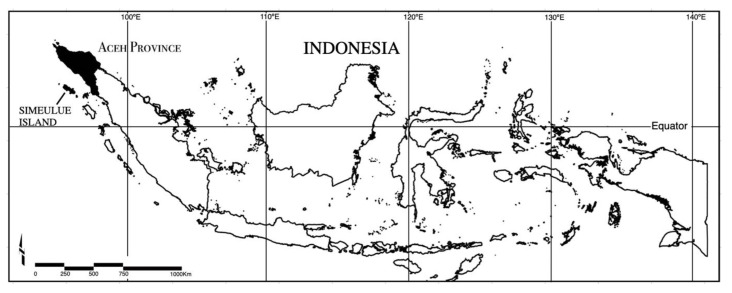
Location of Simeulue Island in Indonesia.

**Figure 2 ijerph-17-07764-f002:**
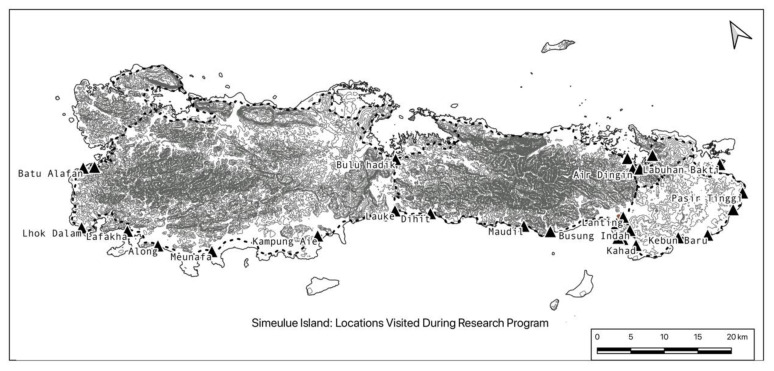
Simeulue Island: Locations Visited During Fieldwork.

**Table 1 ijerph-17-07764-t001:** Context and Gender of Interview Participants.

**Urban**	**Rural**		**Male**		**Female**			
20	17		27		10			
**Age Range (yrs)**	**20–29**	**30–39**	**40–49**	**50–59**	**60–69**	**70–79**	**80–89**	**90+**
**n**	3	12	10	4	4	1	1	2

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
