# Peer review of "Sustaining a Transformative Disaster Risk Reduction Strategy: Grandmothers’ Telling and Singing Tsunami Stories for over 100 Years Saving Lives on Simeulue Island"

_ijerph, 2020, doi:10.3390/ijerph17217764_

Round 1

Reviewer 1 Report

General comments:

In this manuscript, the authors introduce a novel perspective in mitigating the impacts of natural hazard events and demonstrate that grandmothers are the keys of dealing with the situation. In general, the content of the paper is abundant, and the way of dealing with natural hazard events is neoteric for me. However, in my opinion, the logic of the paper is not clear, and the division of paragraphs is not reasonable.

Specific comments:

Line 1: About the title, the semicolon is inappropriate.

Line 32: “attributed to” not “is attributed to”.

Line 137: “smong” appears many times in the above text without explanation.

Line 156: Document citation format should be uniform.

Line 200: The part of introduction is too long and logically confused.

Line 226: The title should be placed above the table.

Line 231: There should be no spaces before the paragraph.

Line 277: The story of Anton is too long, and the part about Anton is not necessary for this paper. There are four terms in table 3 and table 4, so there should be four interviewed parts responding to the above terms.

Line 289: This paragraph should be italicized.

Line 339: The end of this paragraph should be semicolon.

Line 603: “a whole population” is inappropriate.

Line 606: The conclusions part is also too long.

Line 619: “observations” not “observation s”.

Line 710: “l” is extra.

Author Response

Tabulation of reviewer's comments, (final) location in document, author's response and action taken uploaded.

Reviewer 2 Report

1. Section “1.1 The Simeulue Scene: Background to Simeulue and ‘Smong’” should be considered as a separate section, not in the introduction.

2. In the introduction of the work, the authors should clearly state the main contribution or contributions of the work. Many of the viewpoints in this paper come from other literature. I hope the author can clarify the main points and contributions of this paper.

3. 239 Throughout the interviews, grandmothers featured as a primary source of the DRR wisdom that

332 The analysis of the interviews of Orang Simeulue identified six narrative components which,

333 taken together, form a potent system for communicating DRR information

It seems that “grandmothers featured as a primary source of the DRR wisdom” and “six narrative components” are the important results in this paper. The discussion section and results section should be interrelated and mutually supportive. I hope the discussion section is closely related to these results.

4. Conclusions should be clear, concise and structured and should be a summary of the results and discussion.

Author Response

(The authors gave the same response as above.)

Reviewer 3 Report

The paper is generally well written, logically structured and comprehensibly phrased. The length of the manuscript is appropriate.

The experiences and everyday life of those people affected on the ground have so far been investigated and used far too little for Disaster Risk Reduction. Conversely, disasters are particularly serious when people already have to live in vulnerability, politically incapable and without any scope for action of their own. Therefore, I really appreciate that the authors take account of people's cultures, beliefs and attitudes in relation to risk and DRR.

However, the authors are using the complex concept of “community” quite uncritically in their manuscript. „Community“ - or it’s adjunct „community-based" - is one of the most widely used words/concepts in a number of fields of application ranging from the academic world to aid agencies and international organisations – and it is very often used without reflection on its meaning or even a definition. Using “community” avoids examining the heterogeneity of them – these include power relations, divisions, conflicts and oppressions that are in much of the so-called global South a significant part of the reasons why some people are vulnerable. In relation to "local" or "place-bound" communities, a rather one-dimensional and static understanding of community is often used in academia as well as in DRR or CCA interventions, ignoring the social dynamics and the multiple, sometimes contradictory layers of meaning embedded in the word "community".

Meanwhile, the concept of "community" is being critically reflected on in the academic literature but recent debates in this field have not been taken into account by the authors.

Some of the chapters are divided into many very short paragraphs. The authors should attempt to bring together paragraphs which are in a context of meaning. This would also make the text easier to read.

In my view, not all the tables in the Methods chapter are necessary, and in particular the information in Tables 3 and 4 (lines 248 and 256) can just as well be included in the text. Given the number of 37 interviews, absolute figures should be used, as the percentages suggest representativeness.

Only as a suggestion: make the information on the respondents in the chapter Results anonymous.

Author Response

(The authors gave the same response as above.)

Round 2

Reviewer 2 Report

The revision is satisfactory. I suggest this manuscript can be published.